# A secretory kinase complex regulates extracellular protein phosphorylation

**Jixin Cui[1], Junyu Xiao[1†‡], Vincent S Tagliabracci[1], Jianzhong Wen[1§], Meghdad Rahdar[1¶], Jack E Dixon[1,2,3]***

[1]Department of Pharmacology, University of California, San Diego, La Jolla, United States; [2]Department of Cellular and Molecular Medicine, University of California, San Diego, La Jolla, United States; [3]Department of Chemistry and Biochemistry, University of California, San Diego, La Jolla, United States

**Abstract** Although numerous extracellular phosphoproteins have been identified, the protein kinases within the secretory pathway have only recently been discovered, and their regulation is virtually unexplored. Fam20C is the physiological Golgi casein kinase, which phosphorylates many secreted proteins and is critical for proper biomineralization. Fam20A, a Fam20C paralog, is essential for enamel formation, but the biochemical function of Fam20A is unknown. Here we show that Fam20A potentiates Fam20C kinase activity and promotes the phosphorylation of enamel matrix proteins in vitro and in cells. Mechanistically, Fam20A is a pseudokinase that forms a functional complex with Fam20C, and this complex enhances extracellular protein phosphorylation within the secretory pathway. Our findings shed light on the molecular mechanism by which Fam20C and Fam20A collaborate to control enamel formation, and provide the first insight into the regulation of secretory pathway phosphorylation.

*For correspondence: jedixon@ucsd.edu

**Present address:** †State Key Laboratory of Protein and Plant Gene Research, School of Life Sciences, Peking University, Beijing, China; ‡Peking-Tsinghua Center for Life Sciences, Peking University, Beijing, China; §Discovery Bioanalytics, Merck and Co, Rahway, United States; ¶ISIS Pharmaceuticals Inc., Carlsbad, United States

**Competing interests:** The authors declare that no competing interests exist.

## Introduction

Reversible phosphorylation is a fundamental mechanism used to regulate cellular signaling and protein function. In the past several decades, efforts have been focused on understanding phosphorylation events in the cytoplasm and nucleus. However, little is known about the function and regulation of protein phosphorylation within the secretory pathway, despite the fact that numerous secreted proteins that carry out important biological functions are phosphorylated (*Bahl et al., 2008*; *Zhou et al., 2009*; *Carrascal et al., 2010*; *Salih et al., 2010*; *Stone et al., 2011*). This is likely because the kinases in the secretory pathway have only recently been discovered (*Ishikawa et al., 2008*; *Tagliabracci et al., 2012*, *2013*; *Bordoli et al., 2014*).

Our laboratory recently identified a family of secretory pathway kinases, which includes the ortholog of *Drosophila* Four-jointed (Fj) as well as Fam20A, Fam20B, Fam20C, Fam198A and Fam198B (*Ishikawa et al., 2008*; *Tagliabracci et al., 2012*). Fam20C was identified as the long-sought Golgi casein kinase, which phosphorylates secreted proteins within Ser-x-Glu/phospho-Ser (SxE/pS) motifs, including casein and members of the secretory calcium binding phosphoprotein (SCPP) family (*Kawasaki and Weiss, 2008*; *Tagliabracci et al., 2012*). Furthermore, some 75% of phosphoproteins identified in human serum and cerebrospinal fluid contain phosphate within the Fam20C consensus sequence, suggesting that Fam20C may have a broad spectrum of substrates and therefore play a major role in establishing the secreted phosphoproteome (*Bahl et al., 2008*; *Zhou et al., 2009*; *Salvi et al., 2010*; *Tagliabracci et al., 2013*). Nonetheless, how Fam20C is regulated is unknown. In addition to Fam20C, Fj, the founding member of the family, phosphorylates extracellular cadherin domains (*Ishikawa et al., 2008*). Fam20B phosphorylates xylose within a tetrasaccharide linkage region of O-linked proteoglycans and functions to promote

**eLife digest** Some proteins must be modified in order to work effectively. One common modification is to add a phosphate group to the protein, which is performed by enzymes called protein kinases. Although most of the protein kinases work on proteins inside the cell, it was discovered recently that a small group of kinases work within the 'secretory pathway' and modify proteins that are released (or secreted) out of cells.

One such secretory pathway kinase—called Fam20C—phosphorylates a wide range of secreted proteins and helps to ensure the proper development of bones and teeth. Specifically, Fam20C and a closely related protein called Fam20A are important for forming enamel, the hardest substance in human body, which makes up the outer surface of teeth. However, the exact role of Fam20A is unknown.

Cui et al. now show that Fam20A binds to Fam20C, and this increases the ability of Fam20C to phosphorylate the proteins that form the 'matrix' that guides the deposition of the enamel minerals. Furthermore, mutations in Fam20A lead to the inefficient phosphorylation of enamel matrix proteins by Fam20C, and prevent proper enamel formation. The results raise the possibility that similar mechanisms of secretory kinase activation may also be important in other biological processes where many secreted proteins need to be phosphorylated rapidly.

glycosaminoglycan chain elongation (*Koike et al., 2009*; *Wen et al., 2014*). The biochemical functions of the other members of this protein family are unknown. Here we address the function of Fam20A, the closest paralog of Fam20C.

Both Fam20C and Fam20A are implicated in biomineralization. Fam20C deficiency in humans is associated with a severe and often lethal osteosclerotic bone dysplasia known as Raine syndrome (*Simpson et al., 2007*). Patients with non-lethal forms of Raine syndrome exhibit hypophosphatemia, ectopic calcifications and dental anomalies (*Simpson et al., 2009*; *Fradin et al., 2011*; *Rafaelsen et al., 2013*). These patients have misformed dentin and enamel, suggesting that Fam20C is essential for dentinogenesis and amelogenesis (*Fradin et al., 2011*; *Rafaelsen et al., 2013*). Furthermore, patients with *FAM20A* gene mutations also develop disorders of enamel formation referred to as Amelogenesis Imperfecta (AI), which is often accompanied by ectopic calcification, such as nephrocalcinosis (*O'Sullivan et al., 2011*; *Cho et al., 2012*; *Jaureguiberry et al., 2012*; *Cabral et al., 2013*; *Wang et al., 2013a*, *2014*; *Kantaputra et al., 2014a*, *2014b*). Notably, both Fam20A-knockout (KO) and Fam20C-KO mouse models have been generated and they exhibit similar enamel phenotypes, suggesting that Fam20A and Fam20C might function in the same pathway to control enamel formation (*Vogel et al., 2012*; *Wang et al., 2012*, *2013c*).

Further evidence to support a role for extracellular protein phosphorylation in the regulation of enamel formation comes from the discovery that AI can be caused by a missense mutation in the secreted protein enamelin (ENAM) that disrupts phosphorylation within an SxE motif (*Chan et al., 2010*). ENAM is an enamel matrix protein that, together with amelogenin X (AMELX), ameloblastin (AMBN) and amelotin (AMTN), makes up a subfamily of the SCPPs (*Kawasaki and Weiss, 2008*). These proteins are secreted from specialized cells known as ameloblasts and act as a scaffold for enamel calcification (*Hu et al., 2007*; *Moradian-Oldak, 2012*). ENAM, AMELX, AMBN and AMTN are all phosphorylated within SxE motifs and are therefore potential Fam20C substrates (*Kawasaki and Weiss, 2008*). In addition to ENAM phosphorylation, phosphorylation within a conserved SxE motif in AMELX also appears to be important for enamel mineralization (*Kwak et al., 2009*).

Here we show that Fam20C phosphorylates enamel matrix proteins in vitro and in cells. Even though Fam20A lacks a residue critical for catalysis and appears to be a pseudokinase, it forms a functional complex with Fam20C and allosterically activates Fam20C to efficiently phosphorylate secreted proteins. Our results reveal a novel mechanism to regulate secretory protein phosphorylation, and provide a molecular link between an observed biochemical function of Fam20A and the phenotype of patients with AI.

## Results

### Fam20C phosphorylates enamel matrix proteins in vitro and in ameloblasts

To determine whether enamel matrix proteins are substrates of Fam20C, we expressed and purified recombinant human AMELX, AMTN, and the conserved proteolytic fragment of ENAM (aa173–277) found in developing enamel as 6× His tag-fusion proteins in *Escherichia coli* (*Al-Hashimi et al., 2009*). Recombinant Fam20C phosphorylated each of these proteins in a time-dependent manner in vitro, whereas an inactive Fam20C D478A mutant did not (*Figure 1A*). ENAM (173–277) contains two SxE motifs (S191 and S216), both of which are highly conserved and known to be phosphorylated (*Fukae et al., 1996*; *Al-Hashimi et al., 2009*). As mentioned above, a S216L mutation causes AI (*Chan et al., 2010*). We generated ENAM S191A and S216L mutants and tested them as substrates for Fam20C. As shown in *Figure 1B*, both the S191A and S216L ENAM mutants exhibited markedly reduced phosphorylation by Fam20C. Further, mutation of both Ser residues completely abolished Fam20C-dependent phosphorylation (*Figure 1B*). Thus, Fam20C phosphorylates ENAM (173–277) on S191 and S216.

To confirm that Fam20C is responsible for phosphorylating enamel proteins in cells, we disrupted the *Fam20c* gene in mouse ameloblast-like cells (ALCs; [*Nakata et al., 2003*]) by means of Clustered Regularly Interspaced Short Palindromic Repeats (CRISPR)/CRISPR-associated 9 (Cas9) (*Figure 1—figure supplement 1* and *Figure 3—figure supplement 1C*; [*Ran et al., 2013*]). We expressed V5-tagged AMBN in wild-type (WT) and Fam20C-KO ALCs metabolically labeled with [32P] orthophosphate and subsequently analyzed V5-immunoprecipitates from conditioned medium. Disruption of Fam20C completely eliminated AMBN phosphorylation, suggesting that Fam20C is the kinase that phosphorylates enamel matrix proteins in vivo (*Figure 1C*).

### Fam20A lacks a residue critical for kinase activity

Because Fam20A and Fam20C share a high degree of sequence similarity, are expressed in ameloblast cells, and mutations cause similar enamel phenotypes, we anticipated that Fam20A might

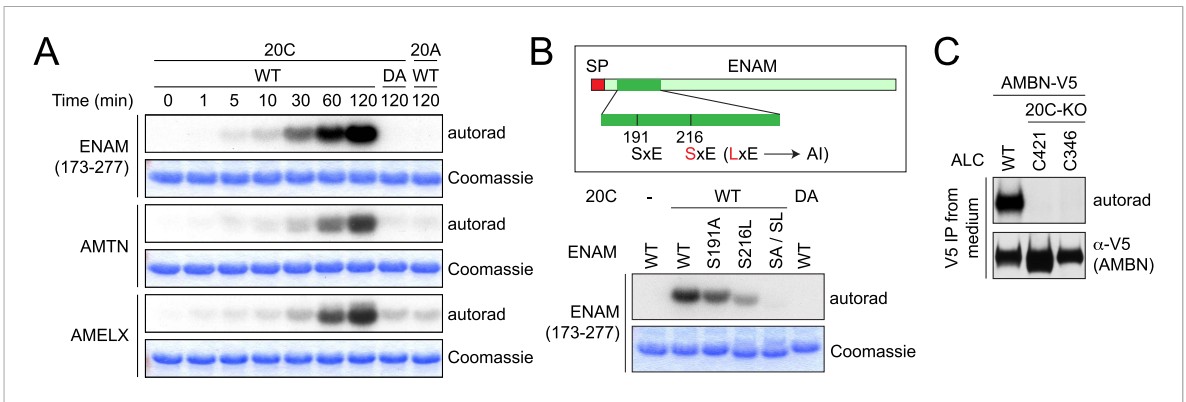

**Figure 1**. Fam20C phosphorylates the enamel matrix proteins within SxE motifs. (**A**) Phosphorylation of enamel matrix proteins by Fam20C in vitro. 250 μg/ml purified recombinant human ENAM (173–277), AMTN or AMELX was incubated with recombinant Fam20C, Fam20C-D478A (DA) or Fam20A (5 μg/ml) at 30˚C in the presence of [γ-32P]-ATP. After reaction, proteins were separated by SDS-PAGE and visualized by Coomassie staining. 32P incorporation was detected by autoradiography. (**B**) Effect of SxE motif mutations on ENAM phosphorylation by Fam20C. Upper: the schematic of ENAM protein. Dark green, ENAM (173–277). SP, signal peptide; AI, Amelogenesis Imperfecta. S216L mutation was reported to cause AI. Lower: phosphorylation of different forms of ENAM (173–277) by Fam20C. SA/SL, S191A and S216L. (**C**) Effect of Fam20C KO on the phosphorylation of AMBN. C-terminal V5-tagged AMBN (AMBN-V5) was expressed in WT ALCs or two Fam20C-KO ALC clones (C421 and C346) that were metabolically labeled with 32P orthophosphate. AMBN-V5 was immunoprecipitated from the conditioned media. Total protein and 32P incorporation were detected by western blot and autoradiography, respectively.

The following figure supplement is available for figure 1:

**Figure supplement 1**. CRISPR-mediated Fam20C KO in ALCs and its effect on osteopontin (OPN) phosphorylation.

also phosphorylate enamel matrix proteins (*Vogel et al., 2012*; *Wang et al., 2013b*). However, recombinant Fam20A was unable to phosphorylate the enamel proteins in vitro (*Figure 1A*). In fact, Fam20A was unable to phosphorylate any protein or carbohydrate substrates we tested, raising the question of whether Fam20A is an active kinase. A number of kinases exhibit intrinsic ATPase activities in the absence of their physiological substrates. Incubation of recombinant Fam20C with ATP resulted in the time-dependent release of phosphate, which correlated with the kinase activity of Fam20C (*Figure 2A*). In contrast, Fam20A was unable to hydrolyze ATP (*Figure 2A*).

ATP binding is known to enhance the thermal stability of kinases, thus increasing the melting temperature ($T_m$) (*Niesen et al., 2007*; *Murphy et al., 2014*). As expected, the $T_m$ of Fam20C significantly increased in the presence of ATP, whereas the $T_m$ of maltose-binding protein (MBP) was unchanged (*Figure 2B*). We also observed an increase in the thermal stability of Fam20A in the presence of ATP, suggesting that Fam20A binds ATP (*Figure 2B*). Collectively, these results support that Fam20A can bind, but not hydrolyze ATP.

To gain insight into the catalytic differences between Fam20A and Fam20C, we looked for cognate residues missing in Fam20A that are critical for Fam20C kinase activity. Most of the residues important for Fam20C activity are conserved in Fam20A, including the metal-binding Asp (D478 in human Fam20C), the catalytic Asp (D458 in human Fam20C), and the ion pair (K285 and E311 in human Fam20C) (*Figure 2C*). However, a conserved Glu in Fam20C (E306 in human), which coordinates the $Mn^{2+}$ ion and the ion-pair Lys and is indispensable for Fam20C kinase activity, is replaced by a Gln (Q258 in human) in all Fam20A orthologs (*Figure 2C,D*, *Figure 2—figure supplement 1* and [*Xiao et al., 2013*]). Because Gln in Fam20A lacks a negatively charged side chain, it may not functionally substitute the Glu in Fam20C.

To assess the effect of this evolutionarily conserved amino acid substitution on the biochemical properties of Fam20A and Fam20C, we generated Fam20A Q258E and Fam20C E306Q mutants. Recombinant Fam20C E306Q exhibited greatly reduced in vitro kinase activity towards casein, ENAM (173–277), as well as osteopontin (OPN), an SCPP and model substrate of Fam20C, as compared to WT Fam20C (*Figure 2—figure supplement 2*). Accordingly, the intrinsic ATPase activity of Fam20C E306Q was abolished (*Figure 2E*). Conversely, Fam20A Q258E was able to hydrolyze ATP and phosphorylate OPN, albeit less robustly than Fam20C (*Figure 2E,F*). These data suggest that Fam20A does not have all the residues essential for catalytic activity and that Fam20A is not an intrinsically active kinase.

## Fam20A stimulates Fam20C-dependent phosphorylation

In order to elucidate the function of Fam20A, we generated Fam20A-KO ALCs by means of CRISPR/Cas9 (*Figure 3—figure supplement 1A*). Depletion of Fam20A had no effect on the expression level of Fam20C (*Figure 3—figure supplement 1C*). Although Fam20A is likely catalytically inactive, we observed that V5-tagged AMBN phosphorylation was greatly diminished in Fam20A-KO ALCs (*Figure 3A*). This effect was not solely restricted to enamel proteins because phosphorylation of V5-tagged OPN was also dramatically reduced in Fam20A-KO ALCs (*Figure 3B*). Reintroduction of Flag-tagged Fam20A in Fam20A-KO ALCs restored the phosphorylation of V5-tagged AMBN and OPN (*Figure 3A,B*). Notably, the phosphorylation of V5-tagged OPN was totally eliminated when the *Fam20c* gene was disrupted in either WT or Fam20A-KO cells (*Figure 3C* and *Figure 3—figure supplement 1B–C*). Furthermore, ectopic expression of Fam20A in the human osteosarcoma cell line U2OS, which produces Fam20C, but not Fam20A, significantly increased phosphorylation of V5-tagged OPN, ENAM and AMBN, without altering Fam20C expression levels (*Figure 3D–F*). In contrast, no phosphorylation of these substrates was detected in Fam20C-KO U2OS cells, even when Flag-tagged Fam20A was overexpressed (*Figure 3D–F* and *Figure 3—figure supplement 2*). Collectively, these results indicate that Fam20A functions to increase Fam20C substrate phosphorylation when Fam20C is present.

## Fam20A increases the activity of Fam20C in vitro

To decipher the mechanism by which Fam20A regulated Fam20C-dependent phosphorylation, we carried out in vitro kinase assays with the combination of recombinant Fam20A and Fam20C. When both proteins were incubated in the reactions, phosphorylation of ENAM, AMTN, or an N-terminal peptide of AMBN fused to a SUMO tag (SUMO-AMBN-N), were all greatly increased (*Figure 4A*).

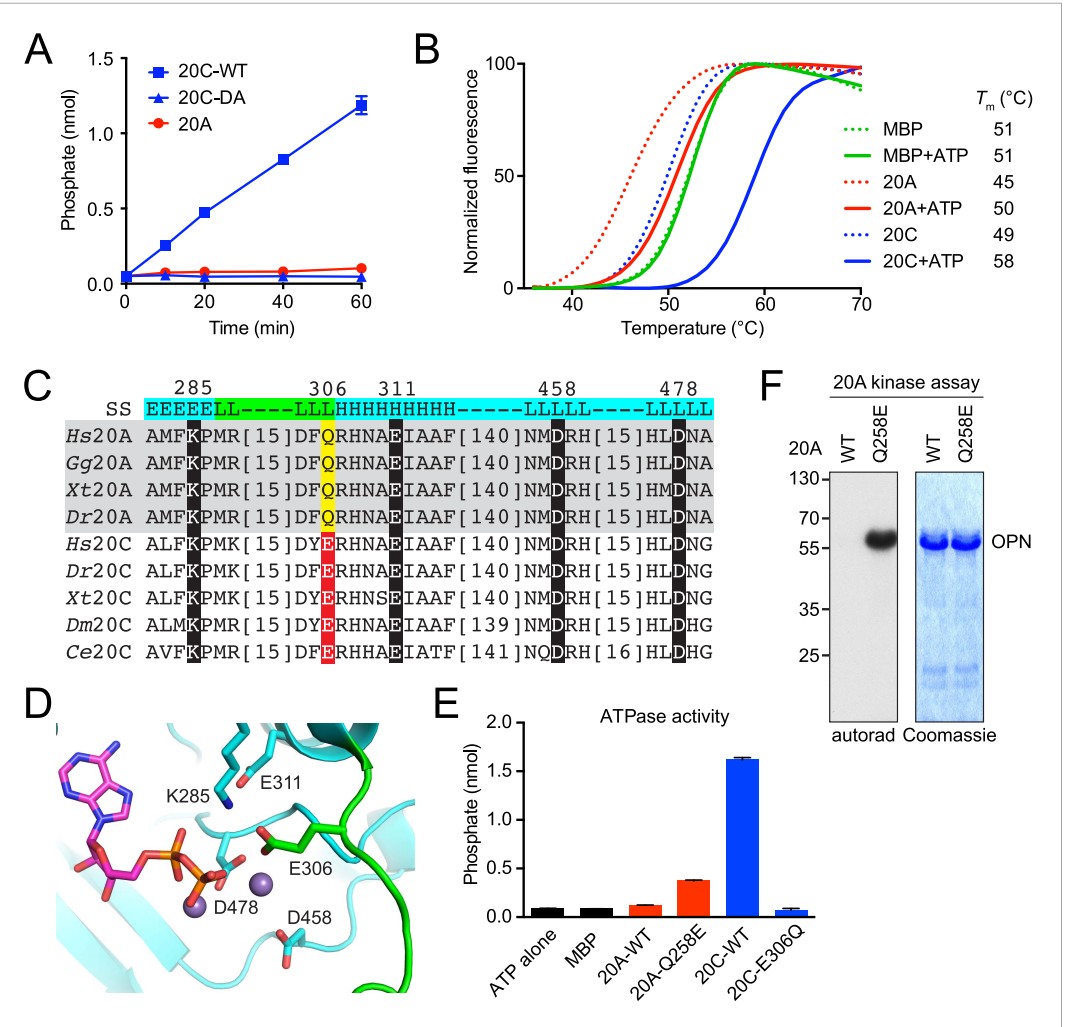

**Figure 2**. Fam20A lacks a residue critical for kinase activity. (**A**) Intrinsic ATPase activity of Fam20C, Fam20C D478A (DA) and Fam20A. Recombinant proteins were incubated with 0.5 mM ATP and 10 mM $MnCl_2$ at 30°C for the indicated time. The amount of phosphate released during incubation was determined using malachite green reagent. (**B**) Thermal stability shift assay of MBP, Fam20A and Fam20C. Protein thermal stability was monitored by the fluorescence generated from binding of the dye SYPRO Orange to the hydrophobic region exposed upon protein denaturation. MBP purified the same way as Fam20A and Fam20C was used as a negative control. Reaction buffer contained 10 mM $MnCl_2$. (**C**) Sequence alignment of Fam20A and Fam20C protein orthologs (*Hs*, *Homo sapiens*; *Gg*, *Gallus gallus*; *Xt*, *Xenopus tropicalis*; *Dr*, *Danio rerio*; *Dm*, *Drosophila melanogaster*; *Ce*, *Caenorhabditis elegans*). Catalytically important residues are highlighted and numbered according to human Fam20C. See *Figure 2—figure supplement 1* for an extended alignment. (**D**) Fam20C active site from PDB:4kqb bound to ADP and $Mn^{2+}$ ions. Conserved canonical kinase ion pair, ion interacting, and catalytic residues are highlighted (cyan sticks) and labeled according to human Fam20C. A Fam20-specific loop is colored in green and contributes a unique active site residue E306. (**E**) Intrinsic ATPase activities of Fam20A Q258E and Fam20C E306Q. Reactions were carried out for 1 hr. (**F**) Effect of Q258E mutation on Fam20A kinase activity. Recombinant OPN was used as the substrate.

The following figure supplements are available for figure 2:

**Figure supplement 1**. Structure-guided sequence alignment of Fam20C-related atypical kinases.

**Figure supplement 2**. Effect of E306Q mutation on Fam20C kinase activity.

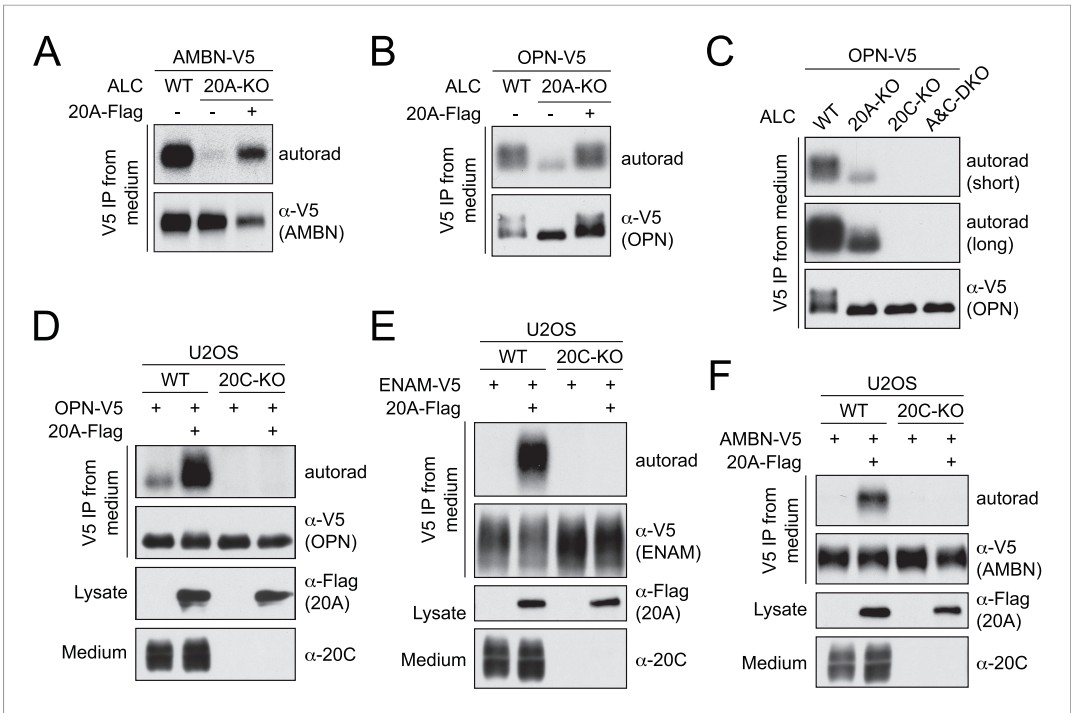

**Figure 3**. Fam20A enhances Fam20C-dependent phosphorylation. (**A** and **B**) Effect of Fam20A KO on the phosphorylation of AMBN (**A**) or OPN (**B**). C-terminal V5-tagged AMBN or OPN (AMBN-V5 or OPN-V5) was expressed in WT or Fam20A-KO ALCs and analyzed as in *Figure 1C*. Flag-tagged Fam20A (20A-Flag) was co-expressed in Fam20A-KO ALCs as indicated. (**C**) Phosphorylation of OPN-V5 in WT, Fam20A-KO, Fam20C-KO, or Fam20A and Fam20C double-KO (A&C-DKO) ALCs. The experiment was performed as in (**B**). Two different exposures of autoradiography are shown. (**D**–**F**) Effect of Fam20A overexpression on the phosphorylation of OPN (**D**), ENAM (173–277) (**E**) or AMBN (**F**) in WT or Fam20C-KO U2OS cells. C-terminal V5-tagged OPN, ENAM (173–277) or AMBN (OPN-V5, ENAM-V5 or AMBN-V5) were expressed in WT or Fam20C-KO U2OS cells and analyzed as in *Figure 1C*. Fam20A-Flag was stably expressed as indicated. Fam20A-Flag in the cell lysates and endogenous Fam20C in the conditioned medium are also shown. Fam20C-KO U2OS cells were generated by means of transcription activator-like effector nuclease (TALEN; *Figure 3—figure supplement 2*).

The following figure supplements are available for figure 3:

**Figure supplement 1**. CRISPR-mediated Fam20A KO and Fam20A/Fam20C double KO (DKO) in ALCs.

**Figure supplement 2**. Transcription activator-like effector nuclease (TALEN)-mediated Fam20C KO in U2OS cells.

---

Recombinant Fam20B did not increase Fam20C-catalyzed ENAM phosphorylation (*Figure 4B*), nor did Fam20A or Fam20C have any effect on Fam20B-catalyzed xylose phosphorylation of the proteoglycan decorin (*Figure 4C*). These results suggest that the activating effect of Fam20A is specific to Fam20C substrate phosphorylation.

To determine whether the phosphorylation sites changed in the presence of Fam20A, we used the ENAM (173–277) S191A and S216L mutants as substrates for in vitro kinase assays. When both Fam20A and Fam20C were included in the reactions, ENAM (173–277) was readily phosphorylated. However, the S191A/S216L double mutant was not, even though it contains 21 additional Ser/Thr/Tyr residues (*Figure 4D*). These results demonstrate that the phosphorylation remains specific to SxE motifs. Further, we replaced each of the Ser and Thr residues with Ala in SUMO-AMBN-N. In vitro kinase assays confirmed that the phosphorylation was specific for SxE in the presence of Fam20A (*Figure 4—figure supplement 1*). As anticipated, the kinase activity of Fam20C was indispensable (*Figure 4D* and *Figure 4—figure supplement 1*). These results point to a model where Fam20A functions as an activator of Fam20C kinase activity.

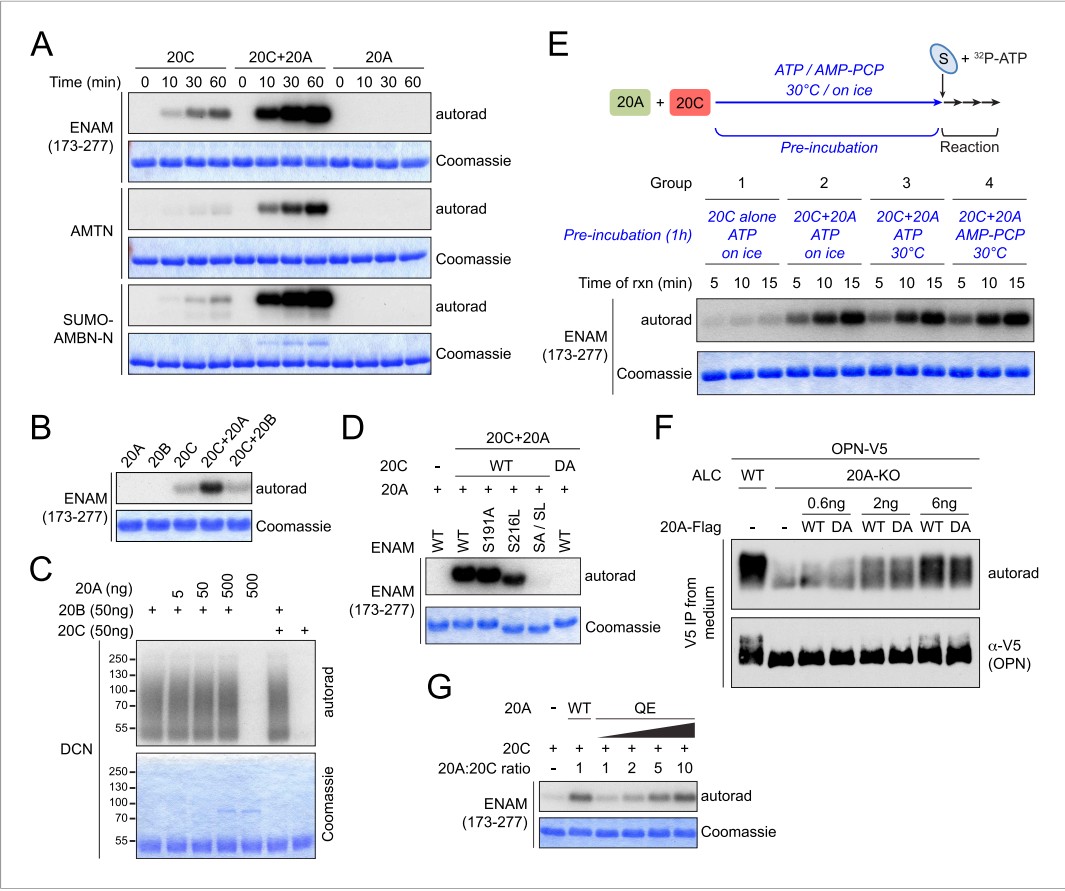

**Figure 4**. Fam20A increases Fam20C kinase activity in vitro. (**A**) Time-dependent $^{32}$P incorporation into enamel matrix proteins by Fam20C, Fam20A, or the combination of Fam20A and Fam20C (molar ratio of 1:1). SUMO-AMBN-N, a N-terminal peptide of AMBN fused to a SUMO tag. (**B**) Effect of Fam20B on ENAM phosphorylation by Fam20C. (**C**) Effect of Fam20A or Fam20C on Fam20B-catalyzed phosphorylation of proteoglycan decorin. Fam20B specifically phosphorylated glycosylated decorin, which migrated as an elongated smear due to the heterogeneity of attached glycosaminoglycans (*Wen et al., 2014*). (**D**) Effect of SxE motif mutations on ENAM phosphorylation by the combination of Fam20A and Fam20C. The experiment was performed as in *Figure 1B*, except that equal amounts of Fam20A and Fam20C were added to the reaction. DA, inactive Fam20C D478A. (**E**) Effect of pre-incubation of Fam20A and Fam20C on ENAM phosphorylation. Fam20A (20 µg/ml) and Fam20C (0.5 µg/ml) were pre-incubated with ATP or nonhydrolyzable AMP-PCP (50 µM), on ice or at 30°C for 1 hr as indicated. Recombinant ENAM (173–277) and excessive ATP were then added to the final concentration of 250 µg/ml and 1 mM, respectively. [γ-$^{32}$P]-ATP was added on the second step to trace ENAM phosphorylation. (**F**) Comparison of the abilities of Fam20A WT and D430A to restore OPN phosphorylation in Fam20A-KO ALCs. The experiment was performed as in *Figure 3B*. For transfection, 0.6–6 ng of 20A-Flag (as indicated) and 2 µg of OPN-V5 DNA were used for each 35 mm dish. (**G**) Effect of Fam20A Q258E (QE) on ENAM phosphorylation by Fam20C in vitro.

The following figure supplements are available for figure 4:

**Figure supplement 1**. Mutagenesis scanning for the phosphorylation site of SUMO-AMBN-N.

**Figure supplement 2**. Effect of Fam20A D430A on ENAM phosphorylation by Fam20C in vitro.

**Figure supplement 3**. Fam20A Q258E does not phosphorylate Fam20C.

To verify this model, we sought to rule out the possibility that Fam20A could be phosphorylated and activated by Fam20C. We pre-incubated Fam20C and excessive Fam20A with ATP on ice or at 30°C prior to the addition of ENAM (173–277) and [γ-$^{32}$P]-ATP, and monitored ENAM phosphorylation. Pre-incubation of Fam20A and Fam20C with ATP at 30°C would allow Fam20C to

phosphorylate Fam20A; however, this did not increase subsequent ENAM phosphorylation when compared to conditions where pre-incubation was carried out on ice (*Figure 4E*, group 2 and 3). Furthermore, ENAM phosphorylation remained constant when Fam20A and Fam20C were pre-incubated with ATP or AMP-PCP, a non-hydrolyzable ATP analogue (*Figure 4E*, group 3 and 4). These findings suggest that the increase of substrate phosphorylation in the presence of Fam20A and Fam20C does not require Fam20C-dependent phosphorylation of Fam20A.

To further confirm that the putative catalytic activity of Fam20A was not required for the increase in Fam20C-dependent substrate phosphorylation, we mutated the putative metal binding Asp of Fam20A (D430A). Although mutation of the corresponding residue in Fam20C eliminated its kinase activity (*Figure 1A*), expression of Flag-tagged Fam20A D430A in Fam20A-KO ALCs increased Fam20C-dependent OPN phosphorylation similar to that of WT Fam20A (*Figure 4F*). Recombinant Fam20A D430A also enhanced ENAM phosphorylation by Fam20C in vitro, although slightly less effectively than WT Fam20A (*Figure 4—figure supplement 2*). Moreover, Fam20A Q258E, which acquires kinase activity, did not phosphorylate Fam20C and was less potent than WT Fam20A to potentiate Fam20C-catalyzed ENAM phosphorylation (*Figure 4G* and *Figure 4—figure supplement 3*). Collectively, our results indicate that Fam20A increases the kinase activity of Fam20C in a catalytically independent manner.

## Fam20A and Fam20C form a complex that is catalytically more active

Because Fam20A enhanced the activity of Fam20C by a mechanism that did not require Fam20A kinase activity, we tested whether Fam20A could directly interact with Fam20C. We incubated recombinant Fam20A with Fam20C and performed gel filtration analysis. Fam20A and Fam20C co-eluted in nearly equal amounts in the higher molecular weight fractions as compared to Fam20A or Fam20C alone, indicating that Fam20A and Fam20C formed a complex of molar ratio of 1:1 (*Figure 5A*). This interaction was specific between Fam20A and Fam20C, because Fam20B and Fam20C did not co-elute during gel filtration (*Figure 5B*). Furthermore, V5-tagged Fam20A, but not V5-tagged Fam20B, could be co-immunoprecipitated with Flag-tagged Fam20C when stably co-expressed in U2OS cells (*Figure 5C*). Thus, Fam20A and Fam20C form a complex in vitro and in the secretory pathway of cells.

Fam20C forms homodimers as judged by its elution profile during gel filtration (*Figure 5A*). Fam20A largely eluted as a monomer; however, the elution profile revealed more heterogeneity, with significant amounts of protein in the fractions consistent with a dimer (*Figure 5A*). Therefore, we ascertained if Fam20A was able to dimerize in cells. Indeed, Flag-tagged Fam20A co-immunoprecipitated with V5-tagged Fam20A, but not V5-tagged Fam20B, when stably co-expressed in U2OS cells, suggesting that Fam20A forms homodimers in cells (*Figure 5D*). Analysis of the Fam20A/Fam20C complex by analytical ultracentrifugation revealed a molecular mass of 245 kDa, which is consistent with a heterotetrameric complex composed of two Fam20A and two Fam20C subunits (*Figure 5E*).

To further understand the effect of Fam20A on Fam20C-catalyzed phosphorylation, we set out to determine the steady-state kinetic parameters of Fam20C in the presence or absence of Fam20A, using ENAM (173–277) S191A as the substrate. When Fam20A was added in excess to the in vitro kinase reaction to maximize complex formation (*Figure 5—figure supplement 1*), the $k_{cat}$ value of Fam20C for ENAM increased by 19-fold (from 0.04 $s^{-1}$ to 0.76 $s^{-1}$), and the $K_m$ for ENAM decreased by ~2.5-fold (from 14.6 μM to 5.6 μM) (*Figure 5F*). This demonstrates that the Fam20A/Fam20C complex catalyzed substrate phosphorylation much more efficiently than Fam20C alone.

Formation of the Fam20A/Fam20C complex raises a possibility that Fam20A associates with the Golgi more tightly than Fam20C and retains Fam20C in the Golgi to phosphorylate more substrates. However, when expressed in mammalian cells, both Fam20A and Fam20C are localized in the Golgi and are secreted into the conditioned medium at comparable levels ([*Tagliabracci et al., 2012*] and *Figure 5G*). We did not observe a redistribution of Flag-tagged Fam20C to the intracellular compartment when co-expressed with V5-tagged Fam20A (*Figure 5G*). These results suggest that Fam20A does not serve as a retention mechanism for the Fam20A/Fam20C complex.

## Fam20A and Fam20C are upregulated and form a complex in the lactating mammary gland

To determine if Fam20A and Fam20C form a complex in vivo, we used lactating mouse mammary gland as a source of endogenous proteins. Fam20C is the *bona fide* kinase that phosphorylates casein, one of the most abundant and highly phosphorylated proteins in milk (*Tagliabracci et al., 2012*).

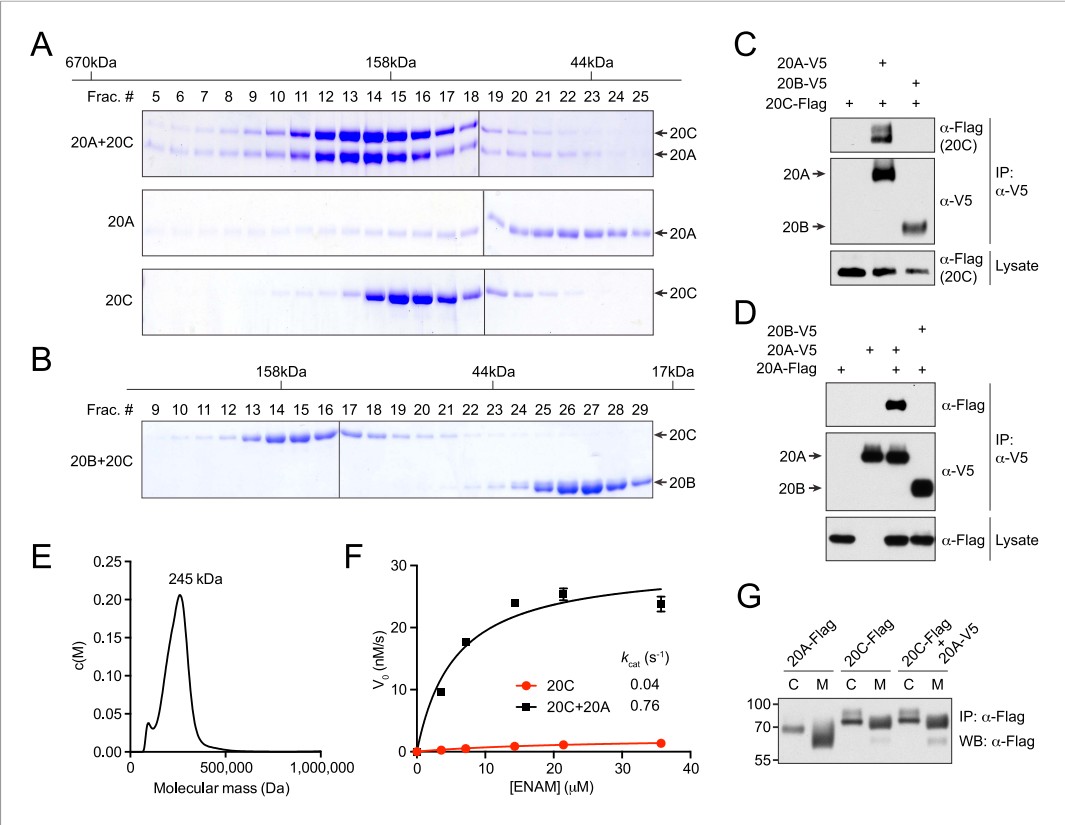

**Figure 5**. Fam20A and Fam20C form a complex that is catalytically more active. (**A**) Gel filtration analysis of Fam20A/Fam20C complex. Purified Fam20A and Fam20C were pre-incubated on ice and passed through Superdex 200. Proteins from different fractions were separated by SDS-PAGE and visualized by Coomassie staining. Fam20A or Fam20C alone were analyzed as controls. The calibration standard is shown on top. (**B**) Gel filtration analysis of Fam20B and Fam20C. (**C**) Co-immunoprecipitation of Fam20A and Fam20C from cells. Fam20C-Flag, Fam20A-V5, or Fam20B-V5 was stably expressed in U2OS cells as indicated. (**D**) Co-immunoprecipitation of Fam20A-Flag and Fam20A-V5 from U2OS cells. (**E**) Analytical ultracentrifugation sedimentation velocity analysis of Fam20A/Fam20C complex. The plot represents the molecular mass distribution c(M) vs the apparent molecular mass (Da). The main peak has a calculated weight of 245 kDa. (**F**) Fam20C kinase reaction initial velocities vs concentration of ENAM (173–277) S191A. Fam20C (2 µg/ml) and Fam20A (40 µg/ml) were used in this experiment. Data points are represented as mean ± SD and fitted by non-linear regression of the Michaelis-Menten equation. (**G**) Immunoprecipitation of Fam20A-Flag or Fam20C-Flag from either cell lysate (C) or the conditioned medium (M). U2OS cells stably expressing Fam20A-Flag, Fam20C-Flag or both Fam20C-Flag and Fam20A-V5 were used.

The following figure supplement is available for figure 5:

**Figure supplement 1**. Saturation of Fam20C kinase activity by Fam20A in vitro.

Fam20C and Fam20A mRNAs in the lactating mammary gland were dramatically elevated as compared to mammary gland from virgin animals, whereas mRNA levels of Fam20B, and other Fj family kinases remained relatively unaffected (*Figure 6A*). Notably, the relative increase in Fam20A and Fam20C mRNAs in lactating mammary gland were virtually identical (*Figure 6A*). In fact, the expression patterns of Fam20A and Fam20C were largely consistent throughout the process of pregnancy and lactation (*Figure 6B*). The mRNA levels of both Fam20A and Fam20C in the mammary gland gradually increased from 13 to 18 days of pregnancy and decreased during lactation and involution, with Fam20A peaking just before and Fam20C peaking just after the birth of the new litter (*Figure 6B*). Consistent with their mRNA levels, Fam20A and Fam20C proteins could only be detected in the mammary glands of lactating, but not virgin mice (*Figure 6C*). Coordinated expression of Fam20A and Fam20C is consistent with the idea that they function together in the mammary gland during pregnancy and

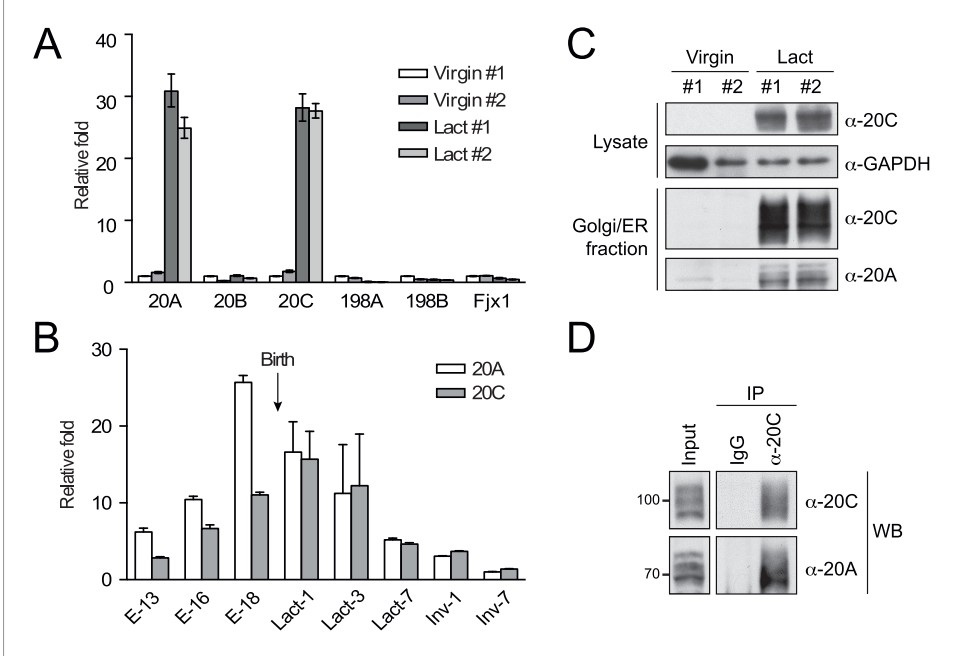

**Figure 6**. Upregulation and complex formation of Fam20A and Fam20C in the lactating mammary gland. (**A**) mRNA levels of Fam20C family members in the mouse mammary gland. The mammary glands from two non-lactating (Virgin) and two 10-day lactating (Lact) mice were isolated and the mRNA levels of Fam20 family members were determined by quantitative (q)RT-PCR. Data are represented as mean ± SD. (**B**) mRNA levels of Fam20A and Fam20C in the mammary glands of pregnant/lactating female mice at different stages. E−13, 16 or 18: embryonic day 13, 16 or 18; Lact-1, 3 or 7: 1, 3 or 7 days after lactation; Inv-1 or 7: 1 or 7 days after involution. The gestation period of mouse is 18–21 days. mRNA levels of Fam20A and Fam20C were determined by qRT-PCR. Data are represented as mean ± SD. (**C**) Protein levels of Fam20A and Fam20C in the mouse mammary glands. Endogenous Fam20A or Fam20C were detected from whole mammary gland lysate or the Golgi/ER-enriched fractions by western blot. (**D**) Co-immunoprecipitation of endogenous Fam20A and Fam20C from Golgi/ER-enriched fractions of the mouse lactating mammary gland. Polyclonal rabbit anti-Fam20C antibody was used for immunoprecipitating endogenous Fam20C. Normal rabbit IgG was used as control for immunoprecipitation.

lactation. Indeed, Fam20A could be co-immunoprecipitated with Fam20C in the Golgi/ER-enriched fraction of the lactating mammary gland (*Figure 6D*). These results demonstrate that Fam20A and Fam20C can form a functional complex in the mammary gland of pregnant/lactating animals.

## Fam20A mutants associated with AI fail to activate Fam20C

Several deletions, truncations and missense mutations in Fam20A cause AI in humans. Because the Fam20A deletions and truncations are likely inactivating, we tested whether the missense Fam20A mutants, L173R, G331D and D403N, could increase Fam20C activity. We co-expressed Flag-tagged Fam20A (WT or AI mutants) and V5-tagged OPN in Fam20A-KO ALCs, and analyzed radiolabeled phosphate incorporation into V5-immunoprecipitates. Expression of WT Fam20A, but not the AI mutants, markedly increased OPN phosphorylation (*Figure 7A*). Moreover, these Fam20A mutants were poorly secreted when ectopically expressed in ameloblast cells (*Figure 7B*), consistent with our previous structural modeling analyses that these mutations likely interfere with the proper folding of Fam20A (*Xiao et al., 2013*).

## Fam20A can increase the activities of Fam20C mutants associated with Raine syndrome

When expressed in U2OS cells, most Fam20C mutants found in Raine syndrome patients could not efficiently phosphorylate OPN. To test whether Fam20A could stimulate kinase activities of Fam20C mutants, we expressed Fam20A with Flag-tagged Fam20C mutants and monitored V5-tagged OPN

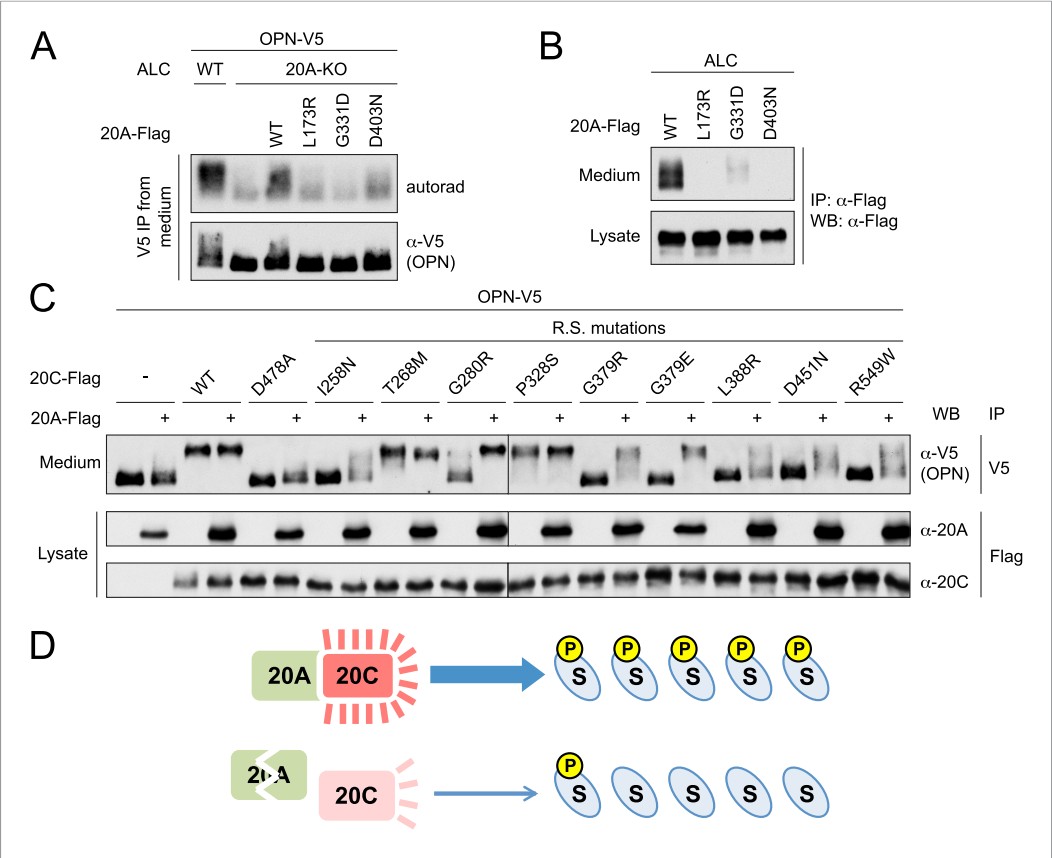

**Figure 7**. Fam20A mutants found in AI patients fail to enhance Fam20C activity, and WT Fam20A is able to increase the activities of Fam20C mutants. (**A**) Effect of Fam20A mutations on OPN phosphorylation. The experiment was performed as in *Figure 3B*. OPN-V5 was expressed in WT or Fam20A-KO ALC. Fam20A-Flag WT, L173R, G331D or D403N was co-expressed as indicated. (**B**) Effect of AI mutations on Fam20A secretion. Fam20A-Flag WT, L173R, G331D or D403N was expressed in ALCs. Fam20A-Flag was immunoprecipitated from conditioned media or cell lysate using anti-Flag antibody and detected by western blot. (**C**) Effect of Fam20A overexpression on Fam20C mutant-catalyzed OPN phosphorylation. OPN-V5 and Fam20C-Flag (WT or mutants as indicated) were expressed in U2OS cells with or without Fam20A-Flag. Secreted OPN-V5 was immunoprecipitated from conditioned media and detected by western blot. OPN phosphorylation was indicated by its mobility shift. D478A is the kinase-dead Fam20C mutation. Other Fam20C mutations were identified from Raine syndrome (R.S.) patients. (**D**) The working model of Fam20A/Fam20C complex. Upper: in the tissues and cells where both Fam20A and Fam20C are expressed, Fam20C forms a complex with Fam20A and exhibits high activity, resulting in high levels of substrate (S) phosphorylation; Lower: when Fam20A is disrupted, Fam20C activity becomes low and substrate phosphorylation is ineffective, leading to diseases like AI.

phosphorylation by its mobility during SDS-PAGE. As expected, WT and non-lethal Fam20C mutants (T268M and P328S), but none of the other mutants induced a mobility shift in OPN (*Figure 7C*), consistent with previous reports (*Tagliabracci et al., 2012, 2014*). However, when Fam20A was co-expressed, we observed enhancement of Fam20C kinase activity in every mutant with the exception of T268M (*Figure 7C*). Because several of the Fam20C mutations are predicted to prevent proper folding/stability, ectopic expression of Fam20A likely promoted not only Fam20C kinase activity, but also Fam20C stability. These results suggest that Fam20A could help stabilize Fam20C mutations and that recapitulating this effect by pharmacological means may be of therapeutic benefit to patients harboring these Fam20C mutations.

## Discussion

In this study, we have shown that Fam20A and Fam20C form a functional complex to phosphorylate secreted proteins within SxE motifs. In the Fam20A/Fam20C complex, Fam20C is catalytically active,

whereas Fam20A functions as an allosteric activator to increase Fam20C kinase activity. Because the residual time of secreted proteins within the secretory pathway is limited, the formation of Fam20A/Fam20C complex, whose turnover number ($k_{cat}$) is dramatically increased, may reflect an important mechanism to achieve high stoichiometry of secreted proteins phosphorylation. Our model suggests that loss-of-function mutations in Fam20A result in inefficient phosphorylation of enamel matrix proteins by Fam20C, which prevents proper enamel formation and leads to AI. The degree of phosphorylation of enamel matrix proteins required for proper enamel formation is unknown. However, patients with hypomorphic mutations in Fam20C, including T268M and P328S, survive infancy but developed dental defects (*Fradin et al., 2011*; *Rafaelsen et al., 2013*), indicating that optimal Fam20C activity is critical for enamel formation.

Our proposed mechanism for activation of Fam20C by Fam20A is supported by the fact that both Fam20A and Fam20C deficiency result in impaired biomineralization in humans and mice. In contrast to Fam20C, which is ubiquitously expressed, Fam20A expression has only been detected in dental tissues, lactating mammary gland, parathyroid gland and kidney (*Vogel et al., 2012*; *Wang et al., 2014*). Further, Fam20A deficiency does not cause complete loss of Fam20C-dependent substrate phosphorylation. These are consistent with the fact that patients harboring Fam20A mutations display mild and localized biomineralization phenotypes, as compared to the more severe forms caused by Fam20C mutations.

It is increasingly apparent that in addition to phosphotranfer, kinase domain-containing proteins can play non-catalytic roles as scaffolds or allosteric regulators (*Zeqiraj and van Aalten, 2010*; *Shaw et al., 2014*). Approximately 10% of the human kinome encodes pseudokinases, which are predicted to lack essential catalytic residues and primarily exert non-catalytic functions (*Manning et al., 2002*; *Zeqiraj and van Aalten, 2010*; *Shaw et al., 2014*). Several pseudokinase-kinase pairs have been revealed, where a pseudokinase binds and potentiates an active kinase. For example, the pseudokinase STRAD, together with MO25, forms a complex with LKB1 and activates LKB1 kinase activity (*Zeqiraj et al., 2009a*, *2009b*). Similarly, the pseudokinase domain of HER3 can dimerize with and allosterically activate catalytically active EGFRs (*Zhang et al., 2006*; *Jura et al., 2009*). Furthermore, the pesudokinase-like inactivated BRAF, generated through binding to ATP-competitive inhibitors or mutation of a residue in the catalytic spine (C-spine), can still allosterically activate other RAF family kinases via dimerization (*Hatzivassiliou et al., 2010*; *Heidorn et al., 2010*; *Poulikakos et al., 2010*; *Hu et al., 2011*, *2013*). Fam20A lacks a negatively charged residue that is critical for kinase activity and therefore is a pseudokinase. Nevertheless, it binds and increases Fam20C activity in a phosphorylation-independent manner. Discovery of the Fam20A-Fam20C pair extends the concept that the activity of a kinase can be allosterically regulated by dimerization with an inactive paralog, which further expands the pseudokinase-kinase mode of regulation to protein phosphorylation within the secretory pathway.

Our result suggests that Fam20A binds ATP, which may stabilize a conformational state that is required for the allosteric function (*Figure 2B*). This mechanism has been reported for other pseudokinases. For instance, the crystal structure of the LKB1-STRAD-MO25 complex revealed that upon binding to ATP and MO25, STRAD was maintained in an active conformation and engaged LKB1 as its 'pseudo-substrate' (*Zeqiraj et al., 2009a*, *2009b*). In agreement with the potential role of ATP in stabilizing Fam20A conformation, Fam20A Q258E, which is able to hydrolyze ATP and therefore will not be locked in an ATP-binding conformation, is less potent than WT Fam20A to activate Fam20C (*Figures 2E, 4G*).

Although dimerization-induced allostery has emerged as a mechanism to regulate protein kinase activity, the interfaces used for mediating kinase domain association appear to be highly diverse (*Lavoie et al., 2014*). The architecture of the Fam20A/Fam20C complex, as well as the detailed mechanism by which interactions between Fam20A and Fam20C increase the kinase activity of Fam20C, are currently unclear. However, the ability of Fam20A to increase the activity of Fam20C mutants identified in patients may lead to a method to allosterically boost Fam20C activity and this may have therapeutic potential. A crystal structure of the Fam20A/Fam20C complex will be of paramount importance.

As mentioned above, in addition to its critical role in biomineralization, Fam20C also appears to be important for maintaining the secreted phosphoproteome. However, regulation of Fam20C-dependent secreted protein phosphorylation is poorly understood. Our data demonstrate that Fam20C activity can be allosterically potentiated by Fam20A, and this increased catalytic activity is critical for enamel formation. Because the expression of Fam20A is tissue-specific, it is unlikely that

Fam20A functions as a global regulator of Fam20C, arguing the existence of additional regulatory mechanisms for Fam20C activity. Indeed, we show that both Fam20A and Fam20C mRNA levels are dramatically increased in the lactating mammary gland, suggesting transcriptional regulation, which may involve pregnancy-related hormones.

In conclusion, our work suggests a model by which Fam20A regulates secretory pathway phosphorylation and tooth enamel formation via activation of Fam20C. We hypothesize that in cells where Fam20A and Fam20C are co-expressed, Fam20A forms a complex with Fam20C and enhances its kinase activity, leading to high levels of substrate phosphorylation (Figure 7D). This would be especially important in the biological processes where large amounts of substrates need to be phosphorylated, such as teeth development or milk production. Conversely, Fam20A deficiency results in reduced Fam20C activity and basal levels of substrate phosphorylation (Figure 7D). Phenotypes, such as enamel defects, will present when the basal phosphorylation is insufficient to support the corresponding biological process. Thus, our findings have uncovered a novel mechanism to regulate secretory pathway phosphorylation and biomineralization.

# Materials and methods

## Cloning

Human cDNAs for FAM20A, FAM20B, FAM20C, ENAM and AMELX were from Open Biosystems. Human AMBN and AMTN cDNAs were from DNASU. For transient expression in mammalian cell, full length AMELX, AMBN or AMTN was cloned into pCDNA4 with a C-terminal V5 tag, and Fam20A, Fam20B or Fam20C was cloned into pCCF with a C-terminal Flag tag. ENAM (173–277) was cloned into pCDNA4 with a N-terminal signal peptide and a C-terminal V5 tag. For stable expression in mammalian cells, Fam20A, Fam20B or Fam20C was cloned into pQCXIP or pQCXIH (Clontech, Mountain View, CA) with a C-terminal Flag or V5 tag as specified. For recombinant expression in *E. coli*, AMBN (27–48) was cloned into pSUMO, and AMELX (17–191), AMTN (17–219) or ENAM (173–277) was cloned into pET28 with a N-terminal 6× His tag. For insect cell expression, human Fam20A (63–529) or Fam20C (93–584) was cloned into a modified pI-secSUMOstar vector (LifeSensors, Malvern, PA), in which the original SUMO tag was replaced by a MBP tag and a tobacco etch virus (TEV) protease site. Site-directed mutagenesis was performed using QuikChange (Agilent Technologies, Santa Clara, CA). All the constructs were verified by DNA sequencing.

## Protein expression and purification

MBP-tagged Fam20A and Fam20C proteins were expressed in Hi5 insect cells and purified from the conditioned medium as described previously (Xiao et al., 2013). The MBP tag was removed using gel filtration chromatography following TEV protease digestion. 6× His-tagged AMELX, AMTN, ENAM and SUMO-AMBN-N were expressed in *E. coli* BL21 (DE3) and purified by Ni-NTA-agarose chromatography (Qiagen, Venlo, Netherlands) as described previously (Tagliabracci et al., 2012). Flag-tagged Fam20C, Fam20B and decorin were expressed in 293T cells and purified from the conditioned medium as described previously (Wen et al., 2014).

## Antibodies

Flag (M2) antibody was purchased from Sigma (St. Louis, MO). V5 antibodies from Millipore (Billerica, MA; AB3792) and Life Technology (Carlsbad, CA; R960-25) were used for immunoprecipitation and immunoblotting, respectively. Rabbit polyclonal anti-Fam20C antibody was raised against and affinity purified with recombinant Flag-tagged Fam20C produced in 293T cells. Rabbit polyclonal anti-Fam20A antibody was raised against and affinity purified with recombinant Fam20A produced in insect cells.

## In vitro kinase assay, ATPase assay and gel filtration analysis

In vitro kinase assays using various Fam20A substrates were performed in a buffer containing 50 mM Hepes (pH 7.0) and 10 mM $MnCl_2$ at 30°C. 100 μM [γ-$^{32}$P]-ATP (specific activity, 500–2000 cpm/pmol) was used in the reaction unless specified. Reactions were terminated by the addition of Laemmli buffer and 20 mM EDTA and boiling for 5 min. Proteins were separated by SDS-PAGE and visualized by Coomassie staining. $^{32}$P incorporation into the substrates was detected by autoradiography.

For kinetic studies using ENAM (173–277) S191A as the substrate, reactions were performed in 50 mM Hepes (pH 7.0), 60 mM NaCl, 10 mM MnCl$_2$, 0.5 mg/ml BSA, 100 μM [γ-$^{32}$P]-ATP (specific activity, 5000 cpm/pmol) and various amount of ENAM (173–277) S191A. Reactions were initiated by the addition of recombinant Fam20C (2 μg/ml) or a combination of Fam20C (2 μg/ml) and Fam20A (40 μg/ml), and incubated for 40 s at 30°C. Proteins were separated by SDS-PAGE and visualized by Coomassie staining. The gel was dried on a piece of filter paper and the ENAM band was cut out for scintillation counting to quantify the incorporated $^{32}$P radioactivity. Data points were fitted by non-linear regression of the Michaelis-Menten equation.

ATPase assay was performed in a buffer containing 10 mM Hepes (pH 7.4), 50 mM NaCl, 10 mM MnCl$_2$, 0.5 mg/ml BSA, 0.5 mM ATP and 50 μg/ml protein at 30°C. Phosphate release was quantified by using a malachite green-based colorimetric assay for inorganic phosphate (*Maehama et al., 2000*).

For gel filtration analysis of the Fam20A/Fam20C complex, purified Fam20A and Fam20C proteins (1 mg/ml each) were incubated on ice for 2 hr and then passed through a Superdex 200 column (GE Healthcare, Piscataway, NJ) with a buffer containing 25 mM Tris–HCl (pH 7.4) and 150 mM NaCl. Fam20A or Fam20C alone, or the combination of Fam20B/Fam20C was analyzed the same way as controls.

## Thermal stability shift assay

For the thermal stability shift assays, proteins were diluted to a final concentration of 2 μM in a buffer containing 10 mM Hepes (pH 7.4), 100 mM NaCl and 10 mM MnCl$_2$. ATP (1 mM) was added as indicated. SYPRO Orange dye (5000× stock, Molecular Probes, Eugene, Oregon, S6650) was added to a final concentration of 5× to trace protein denaturation. Thermal scanning (25–85°C at 1°C/min) was performed using a CFX96 Touch Real-Time PCR Detection System (Bio-Rad, Hercules, CA) with the detection channel of 610–650 nm, and the melting curves were normalized to a 0–100 range. $T_m$ corresponding to the midpoint for the protein unfolding transition was calculated by fitting the increasing phase of the melting curve by nonlinear least-squares regression using sigmoidal equations (GraphPad Prism).

## Analytical ultracentrifugation sedimentation velocity analysis

Sedimentation velocity experiment was performed using a Beckman ProteomeLab XL-I analytical ultracentrifuge. Purified Fam20A/Fam20C complex (1 mg/ml) in 10 mM Hepes, pH 7.5, 100 mM NaCl was spun at 30,000 rpm for 5 hr, and the 280 nm absorbance data were recorded. Data analysis was performed using the SEDFIT software (*Schuck, 2000*).

## Mammalian cell culture, transfection and $^{32}$P orthophosphate metabolic labeling

The ameloblast-like cells (ALCs) were kindly provided by Dr John Bartlett at the Forsyth Institute with the permission of Dr Toshihiro Sugiyama at Akita University, Japan. ALC, U2OS and 293T cells were cultured in Dulbecco's modified Eagle's medium (DMEM; Life Technology) supplemented with 10% FBS (Life Technology) and 100 μg/ml penicillin/streptomycin (Life Technology) at 37°C in a 5% CO$_2$ incubator. Transfection was carried out by using FuGENE-6 (Promega, Madison, WI) following the manufacturers' instructions.

For metabolic radiolabeling experiments, ALCs or U2OS cells were seeded at $5 \times 10^5$ cells per well in 6-well plate; 20 hr later, cells were transfected with 4 μg of plasmid encoding C-terminal V5-tagged AMBN, ENAM (173–277) or OPN. 20 ng, or the amounts specified in *Figure 4D*, of pCCF-Fam20A-Flag was co-transfected, when the effect of Fam20A ectopic expression was ascertained. 2 days after transfection, metabolic labeling was started by replacing the medium with phosphate-free DMEM containing 10% dialyzed FBS and 1 mCi/ml $^{32}$P orthophosphate (PerkinElmer, Waltham, MA). After labeling for 8 hr, the conditioned medium was collected and the cell debris was removed by centrifugation. V5-tagged proteins were immunoprecipitated from the supernatant and analyzed for protein and $^{32}$P incorporation by immunoblotting and autoradiography, respectively.

## Generation of Fam20A/Fam20C-KO cell lines

Disruption of *Fam20a* or *Fam20c* gene in ALCs was carried out by means of CRISPR/Cas9 (*Ran et al., 2013*). The guide sequences targeting exon 1 of mouse *Fam20a* or *Fam20c* gene were designed using

the online tool at crispr.mit.edu and cloned into pX330 (Addgene, Cambridge, MA). To generate Fam20A-KO or Fam20C-KO clones, ALCs were co-transfected with pX330 constructs containing the targeting sequences and pEGFP-C1 (Clontech) at molar ratio of 4:1. 1 day after transfection, cells were trypsinized and single-cell sorted into 96-well plate using BD FACSJazz cell sorter. To screen cell clones with disrupted *Fam20a* or *Fam20c* gene, genomic DNA from expanded single clones was isolated using Quick-gDNA MiniPrep kit (Zymo Research, Irvine, CA). The genomic region flanking the targeting sequence was amplified by PCR and subjected to DNA sequencing. To generate Fam20A/Fam20C double KO cell line, the *Fam20c* gene was disrupted in Fam20A-KO clone A21 using targeting sequence m20c#3 (see below).

The CRISPR targeting sequences used in this study are as follows: m20c #3: 5′-GTGGCGCGTC GGTCCAGCTT-3′ (for clone C346 and AC33); m20c #4: 5′-GGGCTCCCCGGAGGATCGCG-3′ (for clone C421); m20a #3: 5′-CAGGCGGCGGGGCGCTCGCC-3′ (for clone A21). The following primers were used to screen for the disrupted alleles: mouse *Fam20a*, 5′-GGTCCCCAAGTTCAGGGAAG-3′ (forward) and 5′-CTGTGACGGCAGAGTGAAGT-3′ (reverse); mouse *Fam20c*, 5′-CATGAAGATGA TACTGGTGCG-3′ (forward) and 5′-GTCGCTGTTCACATTAAACAG-3′ (reverse).

FAM20C-KO U2OS cell line was generated by means of transcription activator-like effector nuclease (TALEN). A TALEN binding pair was designed to target the exon 1 of human *FAM20C* gene. The genomic recognition sequences for the left and right TALEN arms are GCGCCGGTTCCG CGTGCT (NN-HD-NN-HD-HD-NN-NN-NG-NG-HD-HD-NN-HD-NN-NG-NN-HD-NG) and GGTGGC CTGCGCGCTGC (NN-HD-NI-NN-HD-NN-HD-NN-HD-NI-NN-NN-HD-HD-NI-HD-HD), respectively. TALEN vectors were assembled using the Golden Gate method (Cermak et al., 2011). The cutting activity of the designed TALEN pair was measured by the SURVEYOR assay (Transgenomics, Omaha, NE). To isolate Fam20C-KO clones, U2OS cells were transfected with the TALEN plasmids by electroporation and subjected to limiting dilution. Cell clones were expanded and genomic DNA was isolated using Quick-gDNA MiniPrep kit (Zymo Research). Disruption of *FAM20C* gene was assessed by PCR using primers 5′-GGACCCACACGCCCG-3′ (forward) and 5′-GCAGGATGCGGAGCG-3′ (reverse) and DNA sequencing. Loss of Fam20C expression was confirmed by immunoblotting.

## Mouse mammary gland RNA and protein isolation, fractionation and qRT-PCR

Mammary glands were harvested from virgin or lactating C57BL/6 female mice. For generating the whole tissue lysate, mammary glands were homogenized in a buffer containing 50 mM Tris–HCl (pH 8.0), 150 mM NaCl, 0.5% NP-40, 10% glycerol, 0.5 mM EDTA and protease inhibitors cocktail. To enrich the ER/Golgi, mammary glands were homogenized in HME buffer containing 10 mM Tris–HCl (pH 7.4), 250 mM sucrose, 1 mM EDTA and protease inhibitors cocktail. The homogenate was centrifuged at 1000×$g$ to remove nuclei and unbroken cells. The supernatant was centrifuged at 3000×$g$ to pellet heavy mitochondria. The 3000×$g$ supernatant was then centrifuged at 17,000×$g$ to pellet the ER, Golgi and light mitochondria. This pellet was resuspended in HME buffer as the ER/Golgi-enriched fraction. For co-immunoprecipitation of Fam20A and Fam20C, NaCl and Triton X-100 were added to the ER/Golgi fraction to the final concentrations of 100 mM and 1%, respectively, prior to the addition of rabbit anti-Fam20C antibody. Normal rabbit IgG was used as control. Immunoprecipitates were subjected to a 6% polyacrylamide gel to separate Fam20A from the heavy chain. Fam20A and Fam20C were detected by immunoblotting.

For qRT-PCR analysis, total RNA was isolated from lactating or non-lactating mammary glands using the NucleoSpin RNA kit (MACHEREY-NAGEL, Bethlehem, PA). A panel of total RNA isolated from mammary glands at various pregnant/lactating stages was purchased from Zyagen (San Diego, CA). cDNA was synthesized using the iScript kit (Bio-Rad). qRT-PCR analysis was performed using the Power SYBR Green PCR Master Mix (Applied Biosystems, Waltham, MA) on Applied Biosystems 7500 Real-Time PCR System with primers as follows: Fam20A, 5′-GGCATCATTGACATGGCCGTCTTT-3′ (forward) and 5′-TTCATCCTGGGAATGTCGTCCGAA-3′ (reverse); Fam20B, 5′-TTCCAAATGGCATGCGATGGTCTG-3′ (forward) and 5′-TAACTGTGGTCCGTAGCTTGCACT-3′ (reverse); Fam20C, 5′-TGAAGATGATACTG GTGCGCAGGT-3′ (forward) and 5′-CAACAGCAATGTGCAAAGCGCAAG-3′ (reverse); Fam198A, 5′-TGACTTTCTGCTTCAGGTCCACGA-3′ (forward) and 5′-ATGCTGAAGGTTGCCAGCATTGTC-3′ (reverse); Fam198B, 5′-AAGCAATGATAGCCATTCCTCTG-3′ (forward) and 5′-CCACTCAGGTCTGG GAACC-3′ (reverse); Fjx1, 5′-CATGCCAGGCTGTTTCCTTTCCAA-3′ (forward) and 5′-TCGGATCCAAT

CTCCACACAAGCA-3′ (reverse). The expression level of target genes was normalized to that of GAPDH.

## Acknowledgements

We acknowledge John Bartlett (Forsyth Institute) and Toshihiro Sugiyama (Akita University) for kindly providing the ameloblast-like cells. We thank Sandra Wiley for technical assistance and Lisa Kinch (UT Southwestern Medical Center) for sequence analyses. We also thank Gregory Taylor and Carolyn Worby for carefully reading and editing the manuscript and members of the JED laboratory for insightful discussions.

## Additional information

### Funding

The funders had no role in study design, data collection and interpretation, or the decision to submit the work for publication.

### Author contributions

JC, Conception and design, Acquisition of data, Analysis and interpretation of data, Drafting or revising the article; JX, Conception and design, Drafting or revising the article, Contributed unpublished essential data or reagents; VST, Drafting or revising the article, Contributed unpublished essential data or reagents; JW, MR, Acquisition of data, Contributed unpublished essential data or reagents; JED, Principal investigator, Conception and design, Analysis and interpretation of data, Drafting or revising the article

### Ethics

Animal experimentation: Procedures involving mice were reviewed and approved by the Institutional Animal Care and Use Committee (IACUC) at the UC San Diego (Protocol #S03039).

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
