## [Decision Letter]

Your paper has been favorably reviewed by Tony Hunter (Senior editor and Reviewing editor) and three reviewers, who all felt that this is an important advance both in our understanding of Fam20C function and also the enigmatic function of pseudokinases. The evidence that *FAM20A* is a pseudokinase is persuasive, and the fact that you are able to introduce a mutation that reactivates *FAM20A* as an active kinase is striking, since this has not been possible to demonstrate with many other pseudokinases investigated, such as STRAD.

There is only one experiment that we recommend that you include in a revised version:

The authors show that Fam20A and Fam20C co-elute by size-exclusion chromatography (SEC) consistent with a molecular weight of ∼200 kDa, which they speculate is a 2:2 complex. A relatively straightforward SEC-MALS experiment (or analytic ultracentrifugation), which would yield an absolute mass of the complex, should be performed to clarify this rather important mechanistic point.

I have included some of the reviewers' minor comments that you may wish to address in a revised version.

1) In Figure 2, an alignment of a 13-residue section from various Fam20A and Fam20C kinases is shown. It might be useful to show a more extended alignment along with some other atypical kinases. In the alignment shown in the 2012 Science paper, the conserved “q” and the “e” are in a sequence that was “omitted” from the alignment. The kinase aficionados would probably appreciate a more extensive alignment.

2) The gel filtration experiments are interesting and deserve some more comment. Does it appear that 20C is a dimer and 20A is a monomer? The data also suggest that in this experiment all of the Fam20A/C is in the complex, and little monomeric 20A or C is present. Why does it then require 15X excess of FAM20A to reach full activity as shown in Figure 5–figure supplement 2? One would expect this just to be dependent on the affinity of the interaction.

3) It would be interesting to add to the Discussion some speculation as to what some of the advantages of a system like this would be. Potentially, phosphorylation of proteins in the ER might not be desirable and perhaps assembly of the complex only occurs in the Golgi? This could function to control spatially and temporally when kinase activity is present? Is Fam20A targeted differently than Fam20C?

4) Several of the figures of the paper (Figures 1, 2, 3, 4 and 5) have supplementary panels that go with the figure. In our view many of these panels contain quite important data that could be included in the main figures rather than being in a supplementary figure.

5) The authors may wish to consider expanding slightly their general discussion on pseudokinases activating protein kinases in the Discussion section. The Fam20A-FAM20C system seems to have similarities with other pseudokinase-kinase pairs such as STRAD and LKB1 that are associated with disease that the general readers may not be aware of. Clearly it seems that major roles for pseudokinases are emerging as activators of protein kinases.

6) The authors show that the Fam20A mutation Q258E restores weak catalytic activity to Fam20A. It would be interesting to know whether the mutation has any effect on the ability of Fam20A to serve as an allosteric activator of Fam20C. That is, Fam20A Q258E evidently binds ATP, whereas wild-type Fam20A probably does not. Whether Fam20A binds ATP or not presumably affects the stability of the protein, which in turn could affect (positively, one would think) its activator capability.

I also have a few minor questions of my own that could serve as discussion points:

1) Does Fam20C phosphorylate Fam20A?

2) Does reactivated Fam20A phosphorylate Fam20C? Is Fam20C itself phosphorylated, based on MS analysis, and could this be regulatory?

3) Is Fam20A more tightly anchored in the Golgi than Fam20A, and could it serve as the main retention signal for the Fam20C/Fam20A holoenzyme in cell types that co-express the two proteins?

4) Is the Fam20C/Fam20A association regulated in any way, and could this be used as a means to modulate Fam20C activity?

---

## [Author Response]

*The authors show that Fam20A and Fam20C co-elute by size-exclusion chromatography (SEC) consistent with a molecular weight of ∼200 kDa, which they speculate is a 2:2 complex. A relatively straightforward SEC-MALS experiment (or analytic ultracentrifugation), which would yield an absolute mass of the complex, should be performed to clarify this rather important mechanistic point*.

We determined the molecular mass of purified Fam20A/Fam20C complex to be 245 kDa by means of sedimentation velocity analytical ultracentrifugation. This mass is consistent with a heterotetrameric complex composed of two Fam20A and two Fam20C subunits. This data is shown in Figure 5.

*I have included some of the reviewers' minor comments that you may wish to address in a revised version*.

*1) In*
Figure 2*, an alignment of a 13-residue section from various Fam20A and Fam20C kinases is shown. It might be useful to show a more extended alignment along with some other atypical kinases. In the alignment shown in the 2012 Science paper, the conserved* “*q*” *and the* “*e*” *are in a sequence that was* “*omitted*” *from the alignment. The kinase aficionados would probably appreciate a more extensive alignment*.

We replaced the previous 13-residue alignment with an extended one, which includes Fam20A and Fam20C orthologs with all the residues critical for Fam20C kinase activity highlighted (Figure 2). This new alignment shows that all the catalytic residues are conserved between Fam20A and Fam20C except for the Q/E. We also added an additional alignment, which includes other atypical kinases as Figure 2—figure supplement 1. We did not include the conserved E306 of Fam20C in the alignment shown in the 2012 Science paper, because this E is a unique feature of Fam20C and we did not realize its catalytic importance until we solved the Fam20C structure. Further, this E is localized before the αC helix within a long loop that is also Fam20-specific. To provide the reader with a better understanding about the spatial relationship of this E to the canonical kinase ion pair, ion interacting, and catalytic residues, we added a figure of the Fam20C active site as Figure 2.

*2) The gel filtration experiments are interesting and deserve some more comment. Does it appear that 20C is a dimer and 20A is a monomer? The data also suggest that in this experiment all of the Fam20A/C is in the complex, and little monomeric 20A or C is present. Why does it then require 15X excess of FAM20A to reach full activity as shown in Figure 5–figure supplement 2? One would expect this just to be dependent on the affinity of the interaction*.

Fam20C eluted as a dimer on gel filtration. Although the major peak on gel filtration appears as a monomer, Fam20A exhibits high heterogeneity and can be found in fractions consistent with a dimer. We believe that Fam20A purified from insect cells equilibrates between monomer and dimer formats, and Fam20C likely stabilizes the Fam20A dimer, which in turn forms the Fam20A/C tetramer. We incorporated these comments in the Results section. We also tested whether or not Fam20A could dimerize in the secretory pathway. We stably co-expressed Flag-tagged and V5-tagged Fam20A in mammalian cells and found that they co-immunoprecipitated from detergent soluble extracts. These results suggest that Fam20A can form a dimer in cells. These results are shown in Figure 5.

We would like to clarify that not all the Fam20A and Fam20C are in the tetramer on gel filtration. We noticed that the UV trace of Fam20A/C complex was slightly asymmetric with a lag on the lower molecular weight side, indicating heterogeneity and partial dissociation of the complex. Consistent with the gel filtration result, the Fam20A/C complex curve of the analytical ultracentrifugation assay also show a lag towards lower molecular mass (Figure 5). Notably, the concentration of Fam20C used in the kinetic assay is 500 to 1000 fold lower than that in the gel filtration assay, which would result in more complex dissociation. Also, Fam20A purified from insect cells likely switches between monomer and dimer without Fam20C, which makes Fam20A/C complex formation even more difficult at low protein concentration. It might also take a longer time for the complex formation to reach equilibration. Thus, excessive amount of Fam20A would be beneficial for the complex formation under the kinetic assay condition.

*3) It would be interesting to add to the Discussion some speculation as to what some of the advantages of a system like this would be. Potentially, phosphorylation of proteins in the ER might not be desirable and perhaps assembly of the complex only occurs in the Golgi? This could function to control spatially and temporally when kinase activity is present? Is Fam20A targeted differently than Fam20C*?

We added some speculation regarding the potential benefits of the Fam20A-Fam20C system. We speculate that the major function of Fam20A is to boost the phosphorylation capacity of Fam20C within the secretory pathway. As shown in Figure 5, Fam20A increases the turnover number (*k*_cat_) of Fam20C by almost 20 fold. Because proteins destined to secretion only reside for certain amount of time within the secretory pathway, higher turnover number enables the Fam20A/Fam20C complex to catalyze more phosphorylation reactions before the substrates are secreted outside the cell. This regulatory mechanism will be advantageous during the processes like enamel formation and lactation, when lots of secreted proteins need to be phosphorylated to high stoichiometry.

We notice that unlike Fam20C, which is ubiquitously expressed, the expression of Fam20A is tissue-specific and perhaps highly dynamic. Thus, controlling the transcription of Fam20A appears to be a mechanism to spatially and temporally regulate Fam20C kinase activity.

Fam20A does not appear to be targeted differently than Fam20C. Our previous results indicate that similar to Fam20C, Fam20A is predominantly localized in the Golgi ([42], Science). Further, as shown in Figure 5, secretion of Fam20A is as efficient as Fam20C and co-expression of Fam20A has little effect on secretion of Fam20C. It is therefore unlikely that Fam20A serves as a major mechanism to redistribute Fam20C activity subcellularly.

*4) Several of the figures of the paper (*Figures 1, 2, 3, 4 and 5*) have supplementary panels that go with the figure. In our view many of these panels contain quite important data that could be included in the main figures rather than being in a supplementary figure*.

Figures 2, 4 and 5 in the revised manuscript were moved from the figure supplements of the previous manuscript.

*5) The authors may wish to consider expanding slightly their general discussion on pseudokinases activating protein kinases in the Discussion section. The Fam20A-FAM20C system seems to have similarities with other pseudokinase-kinase pairs such as STRAD and LKB1 that are associated with disease that the general readers may not be aware of. Clearly it seems that major roles for pseudokinases are emerging as activators of protein kinases*.

We have incorporated discussions about pseudokinase-kinase pairs in the revised manuscript.

*6) The authors show that the Fam20A mutation Q258E restores weak catalytic activity to Fam20A. It would be interesting to know whether the mutation has any effect on the ability of Fam20A to serve as an allosteric activator of Fam20C. That is, Fam20A Q258E evidently binds ATP, whereas wild-type Fam20A probably does not. Whether Fam20A binds ATP or not presumably affects the stability of the protein, which in turn could affect (positively, one would think) its activator capability*.

As shown in Figure 2, the thermal stability of Fam20A increased in the presence of ATP, indicating that Fam20A could bind ATP. We speculate that binding to ATP may stabilize a conformational state of Fam20A that is preferred for the allosteric activation of Fam20C. This mechanism has been reported for other pseudokinases, including STRAD. In the crystal structure of the LKB1-STRAD-MO25 complex, STRAD adopts an active conformation by binding to ATP and MO25, and engages LKB1 as its ‘pseudo-substrate’. In agreement with our hypothesis, Fam20A Q258E, which is able to hydrolyze ATP and therefore will not be locked in an ATP-binding conformation, is less potent than WT Fam20A to activate Fam20C (Figures 2 and 4). Our speculation about Fam20A binding ATP is stated in the Discussion section.

*I also have a few minor questions of my own that could serve as discussion points*:

*1) Does Fam20C phosphorylate Fam20A*?

When relative high concentration of purified Fam20A and Fam20C were incubated in vitro with ATP and Mn^2+^, Fam20A could be phosphorylated. Whether Fam20C also phosphorylates Fam20A under physiological conditions has not been studied. Also, the functional significance of Fam20A phosphorylation is unclear. As demonstrated by Figure 4, increase of Fam20C activity does not require Fam20A being phosphorylated. Although the potential function of Fam20A phosphorylation by Fam20C is an interesting topic, we think it falls beyond the scope of this manuscript.

*2) Does reactivated Fam20A phosphorylate Fam20C? Is Fam20C itself phosphorylated, based on MS analysis, and could this be regulatory*?

As shown in Figure 4—figure supplement 3, reactivated Fam20A does not phosphorylate Fam20C. Phosphorylation of Fam20C has been discussed in another manuscript, which is currently under consideration of another journal. In brief, Fam20C undergoes autophosphorylation. But mutations of the SxE motifs within Fam20C do not appear to affect Fam20C activity.

*3) Is Fam20A more tightly anchored in the Golgi than Fam20A, and could it serve as the main retention signal for the Fam20C/Fam20A holoenzyme in cell types that co-express the two proteins*?

We addressed this question in Figure 5. When stably expressed in mammalian cells, Fam20A and Fam20C are secreted into the conditioned medium at comparable levels, indicating that Fam20A is not primarily retained within the Golgi. Also, we did not observe a redistribution of Flag-tagged Fam20C to the intracellular compartment when co-expressed with V5-tagged Fam20A. These results suggest that Fam20A does not serve as a retention mechanism for the Fam20A/Fam20C complex.

*4) Is the Fam20C/Fam20A association regulated in any way, and could this be used as a means to modulate Fam20C activity*?

The formation of Fam20A/Fam20C complex is at least regulated on the transcriptional level. Fam20C is ubiquitously expressed; the expression of Fam20A is tissue-specific and perhaps highly dynamic given the large increase in Fam20A expression during lactation (Figure 6). Thus, controlling the transcription of Fam20A appears to be a mechanism to spatially and temporally regulate Fam20A/Fam20C complex formation and therefore Fam20C kinase activity. It is also possible that Fam20A/Fam20C complex formation is regulated post-translationally. Although currently we do not have evidence to support such post-translational regulations, it would be an intriguing direction to pursue in the future.